# The evolutionary behavior of chromophoric brown carbon during ozone aging of fine particles from biomass burning

Xingjun Fan[1,3], Tao Cao[2,4], Xufang Yu[1], Yan Wang[1], Xin Xiao[1], Feiyue Li[1,3], Yue Xie[1], Wenchao Ji[1], Jianzhong Song[2], Ping'an Peng[2]

[1]College of Resource and Environment, Anhui Science and Technology University, Fengyang 233100, P. R. China

[2]State Key Laboratory of Organic Geochemistry, Guangzhou Institute of Geochemistry, Chinese Academy of Sciences, Guangzhou 510640, P. R. China

[3]Anhui Province Key Laboratory of Biochar and Cropland Pollution Prevention, Bengbu 233400, P. R. China

[4]University of Chinese Academy of Sciences, Beijing 100049, P. R. China

**Correspondences:** Jianzhong Song (songjzh@gig.ac.cn), Xingjun Fan (fanxj@ahstu.edu.cn)

**Abstract**

Biomass burning (BB) emits large amounts of brown carbon (BrC), however, the evolutionary behavior of BrC in BB emissions (BB-BrC) resulted from complex atmospheric processes are poorly understood. In this study, the transformation of contents and the chromophoric characteristics of BrC in smoke particles emitted by the burning of rice straw (RS), corn straw (CS) and pine wood (PW) under $O_3$ aging are investigated. The $O_3$ aging induced the reduction of light absorption and fluorescence for the BB-BrC, suggesting the decomposition of chromophores and fluorophores. These changes were accompanied by a decrease of aromaticity, average molecular weight and the light absorption capacity for the chromophores, and an increase of humification for the fluorophores. The excitation emission matrix combined with a parallel factor analysis revealed that protein-like components (C3) were predominantly decomposed by $O_3$ aging, while the relative distribution of a humic-like component with highly oxygenated chromophores (C4) gradually increased. In general, the humic-like substances (C1+C2+C4) were transformed to be the most abundant fluorophores for all the BB-BrC samples, which accounted for 84%–87% of the total fluorophores in final $O_3$-aged BB-BrC. Two dimensional correlation spectroscopy (2D-COS) was performed on the synchronous fluorescence, which suggested that the RS- and CS- BrC exhibits the same susceptible fluorophores changes upon $O_3$ aging. It showed that $O_3$ firstly reacted with protein-like fractions (263–289 nm), and then with fulvic-like fractions (333–340 nm). In comparison, the changing sequence of susceptible fluorophores in the PW-BrC to $O_3$ were in the order of fulvic-like with shorter wavelengths (309 nm) > protein-like fluorophores (276 nm) > fulvic-like fluorophores with longer wavelengths (358 nm). The 2D-FTIR-COS analysis showed conjugated C=O and aromatic C=C and C=O groups were the most susceptible functional groups to $O_3$ aging for all BB-BrC. Moreover, it also revealed a consistent sequential change, which is in the order of aromatic OH, conjugated C=O groups and aromatic C=O, aromatic $COO^-$, and finally lignin-derived C-C, C-H and C-O groups. Our results provide new insights into the evolutionary behavior of the chromophoric and fluorescent properties of BB-BrC during $O_3$ aging, which are of great significance for better understanding the heterogeneous oxidation pathways of BB-derived BrC in atmospheric environment.

**Keywords:** Brown carbon, biomass burning, ozone aging, EEM-PARAFAC, 2D-COS

## 1. Introduction

Brown carbon (BrC), as a fraction of organic aerosols with effective light absorption properties, has been a research hotspot in the atmospheric carbonaceous aerosols field (Andreae and Gelencser, 2006; Laskin et al., 2015; Yan et al., 2018). BrC strongly absorbs solar radiation at short ultraviolet-visible wavelengths, which has been estimated to contribute 6%−28% of the total atmospheric aerosol absorption (Chen et al., 2018; Chung et al., 2012; Feng et al., 2013; Liu et al., 2015). Owing to its significant effect on light absorption, BrC can affect both direct radiative forcing and atmospheric photochemistry (Mok et al., 2016; Jo et al., 2016). In addition, BrC may have an adverse effect on human health due to its capacity of enhancing the oxidative potential of atmospheric aerosols (Chen et al., 2019; Ma et al., 2018).

Atmospheric BrC arises from multiple sources, including biomass burning (BB), coal combustion, and secondary atmospheric chemical reactions (Bond, 2004; Chen et al., 2018; Fan et al., 2018; Hecobian et al., 2010; Saleh et al., 2013; Wang et al., 2018). Among them, BB is believed to be one of the important sources (Chen and Bond, 2010; Fan et al., 2016, 2018; Huo et al., 2018;Park and Yu, 2016). Recently, primary BB-BrC has been widely studied, which has provided deep insights into its BB source-dependent chemical and optical properties (Fan et al., 2016, 2018; Park and Yu, 2016). Moreover, both laboratory and field studies have demonstrated that atmospheric aging processes will lead to a transformation of the chemical composition and optical properties of BB-BrC once they are emitted into atmosphere (Fan et al., 2019; Pillar et al., 2014, 2015, 2017; Saleh et al., 2013; Schnitzler and Abbatt, 2018; Sumlin et al., 2017; Wong et al., 2017, 2018; Zhong and Jang, 2014). The relevant processes mainly occur in either atmospheric aqueous phase (Fan et al., 2019; Wong et al., 2017, 2018; Zhao et al., 2015), gas phase (Schnitzler and Abbatt, 2018; Sumlin et al., 2017; Zhong and Jang, 2014), air-particle interface (Pillar-Little and Guzma, 2018 and references therein) with homogeneous or heterogeneous oxidation.

Most studies have investigated BB-BrC aging processes with a focus on aqueous phase oxidations, such as photochemistry (Chang and Thompson, 2010; Zhao et al., 2015), dark OH-induced oxidation (Santos et al., 2016a,b; Santos and Duarte, 2015), and carbonyl compounds reactions with amines/ammonium sulfate (AS) (De Haan et al., 2017; Tang et al., 2016). The oxidation mechanisms and evolution of the optical properties of individual water soluble organic compounds derived from BB emissions, such as levoglucosan (Holmes and Petrucci, 2006), aromatic acids (Santos et al., 2016b), phenolic compounds (Lavis et al., 2017), nitrophenols (Hems and Abbatt, 2018; Lin et al., 2015), methylglyoxal (De Haan et al., 2017) and some other unsaturated carboxylic acids (Gallimore et al., 2011; Lee and Chan, 2007) have been well illustrated. The results suggest that the chromophoric characteristics of aged BrC are highly variable across different precursors and oxidation conditions. Many recent laboratory simulations studies have also tentatively explored the evolution and oxidation mechanisms of water-soluble BB-BrC in aqueous-phase with or without light (Fan et al., 2019; Wong et al., 2017, 2018). These studies have observed that BB-BrC would

undergo both enhancement and bleaching, but a significant bleaching of BrC dominantly occurred over a long aging time. However, these results are far from definitively pronouncing the evolutionary behavior of BB-BrC, mainly because of the high complexity of BB emission (i.e. non-/polar organic compounds, soot particles) and atmospheric oxidants (i.e. $O_3$, $NO_2$, OH radicals).

The heterogeneous oxidations are also important pathways to form or bleach BrC, which have been widely studied for the aging of soot and individual organic compounds by OH, $O_3$, NOx, $SO_2$, HONO (Baduel et al., 2011; Gallimore et al., 2011; He et al., 2017; Li et al., 2013, 2015; Li et al., 2017; Pillar-Little and Guzman, 2018 and references therein). Among them, the $O_3$ aging simulations have been widely conducted, mainly because of $O_3$ is not only a significant pollutant but also an important oxidant in the atmosphere (Chapleski et al., 2016; Chi et al., 2018; Li et al., 2018). The $O_3$ oxidation of soot particles generally change their physicochemical properties (i.e. water solubility, hygroscopicity, size, oxidative potential, and morphology), as well as form new chromophoric compounds (Decesari et al., 2002; Li et al., 2013; Zhu et al., 2019). In addition, the $O_3$ oxidation of organic compounds, such as humic-like substances (HULIS), catechol and its substituted ones, commonly lead to competitive fragmentation and functionalization of chromophores (Baduel et al., 2011; D'Anna et al., 2009; Pillar et al., 2014, 2015, 2017; Sun et al., 2019). To date, however, there has been no systematic investigation on the evolution behaviors of bulk BrC, especially in terms of its chromophoric characteristics, via heterogeneous $O_3$ oxidation of complex BB emissions.

The objective of this study is to comprehensively investigate the evolutionary behavior of fine particulate chromophoric BrC produced by BB during the $O_3$ aging process. For this purpose, the $O_3$ aging of three types of BB smoke particles emitted from rice straw (RS), corn straw (CS) and pine wood (PW) are conducted in a reactor. Then, the levels and chromophoric characteristics of fresh and aged BrC are determined with a total organic carbon (TOC) analysis, ultraviolet-visible (UV-vis) spectroscopy, an excitation-emission matrix (EEM) fluorescence spectroscopy, and Fourier transform infrared (FTIR) spectroscopy. Furthermore, the $O_3$ aging mechanism of BB-BrC is explored using EEM combined with a parallel factor analysis (PARAFAC) and a two-dimensional correlation analysis (2D-COS) applied on the synchronous fluorescence spectra and FTIR. These approaches have allowed us to investigate the levels and types of independent underlying fluorophores of BrC (Chen et al., 2016b;Fan et al., 2019) and explore the dynamic spectral behaviors of chromophores and molecular structures of BrC (Fan et al., 2019;Duarte et al., 2015). Our results have great importance in terms of furthering source apportionment of atmospheric BrC and of improving the accuracy of predicting their effects in atmospheric environments and on climate change.

## 2. Experimental section

2.1. Preparation of the BB smoke particles

The BB smoke particles were prepared in a laboratory sampling system, which was described in detail in our previous study (Fan et al., 2018). The burning experiments were conducted without any control conditions, which mostly like a natural BB process. Briefly, small pieces of RS, CS and PW biomass materials were placed on stainless steel mesh in a combustor and then ignited with an electronic gas lighter. The resulting smoke was introduced into a mixing chamber with diluting air, and the fine smoke particles were collected on quartz filters (Whatman, Ø 90 mm) using a $PM_{2.5}$ sampler (Tianhong Intelligent Instrument Plant, Wuhan, China) at a flow rate of 80 L/min. Finally, BB particles were obtained from each of the three fuels.

## 2.2. Ozone aging of BB smoke particles

The $O_3$ aging experiments of the BB smoke samples were conducted in a self-designed reactor. A schematic diagram of the ozone aging reactor is illustrated in Figure 1. The construction of the reactor and its operation are described in detail in Section S1 of the supporting information (SI). For $O_3$ aging, the sampled BB smoke particle filters were first cut into pieces (area of each piece=1.5 cm$^2$), and then uniformly spread in a glass dish ($\Phi = 90$ mm) before being exposed in an $O_3$ environment in the principal reactor. During the aging process, the temperature, relative humidity (RH), and $O_3$ concentration in the principal reactor were maintained at 25 ºC, 40% and 70 ppm, respectively. To avoid any photolysis of $O_3$ and any photo-chemical reactions, all components in the reactor were wrapped with aluminum foil. Additionally, control experiments were conducted, but no variations of the chromophoric BrC were observed when there was no ozone introduced (Section S2 of SI, Figures S1–S2).

It should be noted that some gas-phase artifacts (i.e. semivolatile and intermediate volatility organic compounds) possibly adsorbed on the quartz filters during sampling (Geller et al., 2006; Parshintsev et al. 2011). However, the amounts of these adsorbed organic artifacts on the filters are very small (0.48-0.98 µgC/cm$^2$) (Arhami et al., 2006; Subramanian et al., 2004), which are much lower than the amounts of the OC fraction (~250-750 µgC/cm$^2$) in fresh BB smoke samples. Therefore, the potential contributions from $O_3$ oxidation of gas-phase artifacts on filters to BB-BrC can be neglected.

[Figure 1]

## 2.3. Water-soluble BrC extraction

At each designed exposure time (0, 1, 2, 4, 8, 12, and 24 h), three pieces of smoke filter for each BB sample were taken out and immersed in ultrapure water. After extracting with sonication for 30 min, the extracts were filtered through a 0.22 µm polytetrafluoroethylene syringe filter (Jinteng, Tianjin, China). The filtrates obtained were designated as water soluble BrC.

## 2.4. Analytical techniques

The organic carbon (OC) content of the fresh and aged water-soluble BB-BrC samples were measured using a total organic carbon analyzer (TOC-VCPN, Shimadzu, Kyoto, Japan). The UV–vis absorption spectra were obtained using a spectrophotometer (UV-2600, Shimadzu, Kyoto, Japan) at wavelengths ranging from 200 nm to 700 nm. The synchronous fluorescence

(SF) spectra were measured using a F-4600 fluorescence spectrometer (Hitachi, Tokyo, Japan) at excitation wavelengths ranging from 200 to 400 nm with a constant offset ($\Delta\lambda = 60$ nm). The EEM spectra were also recorded using the F-4600 fluorescence spectrometer, and the scanning ranges were set to 220–400 nm for excitation and 290–520 nm for emission, with a 5-nm interval. The scanning speed was set to 12000 nm/min. Ultrapure water was used as

blanks to correct the sample UV-vis, SF and EEM spectra. In addition, portions of the sample solutions were freeze-dried before the FTIR analysis. The FTIR spectra (4000–400 cm$^{-1}$) were recorded at room temperature using a Nicolet iS10 FTIR spectrometer (Thermo, USA), and each spectrum was obtained after 64 scans with 4 cm$^{-1}$ resolution. To describe the aging behaviors of the BB-BrC, some useful optical parameters, including the specific UV

absorbance at 254 nm (SUVA$_{254}$) (Fan et al., 2016), the absorption Angstrom exponent (AAE) (Fan et al., 2016; Cheng et al., 2016; Huo et al., 2018; Park and Yu, 2016), mass absorption efficiencies (MAE$_{365}$) (Chen et al., 2016a; Fan et al., 2016; Huo et al., 2018; Park and Yu, 2016) and humification index (HIX) (Qin et al., 2018), were comparatively analyzed. Their calculation protocols are described in detail in Section S3 of the SI.

2.5. Parallel factor analysis

The PARAFAC analysis was performed using MATLAB R2014a (MathWorks, USA) with the DOMFluor toolbox following the procedure introduced by Stedmon and Bro (Stedmon and Bro, 2008). The PARAFAC was computed using 2–7 component models with non-negativity

constraints, and a four-component model was split-half validated on both halves using 63 distinct EEMs for all BB-BrC samples. It is noted that all EEMs were normalized to the area under the Milli-Q water Raman peak (Ex = 350 nm, Em = 365–430 nm), which were collected on the same day to produce corrected fluorescence intensities in Raman units (R.U.) (Lawaetz and Stedmon, 2009). Furthermore, the relative levels of the individual fluorophores

were evaluated using the corresponding maximum fluorescence intensity ($F_{max}$) (Zhang et al., 2014; Matos et al., 2015).

2.6. 2D-COS

The SF spectra (200–400 nm) and FTIR spectra (1800–1000 cm$^{-1}$) of each type of BrC were

analyzed using 2D-COS with the oxidation time as the perturbation variable. The calculations were performed using the 2Dshige software (Shigeaki Morita, Kwansei-Gakuin University, Japan, 2004–2005). Details of the algorithm adopted in the software can be found in previous studies (Duarte et al., 2015; Jin et al., 2018; Hur and Lee, 2014). Finally, the synchronous and asynchronous maps were plotted by using the OriginPro 2018 C software (OriginLab, USA).

**3. Results and discussion**

3.1. Changes of water soluble BrC levels in BB smoke particles

Changes of the water soluble organic carbons (WSOC) content of BrC, calculated as $WSOC_{[t]}/WSOC_{[0]}$, for three types of BB smoke particles during the $O_3$ aging process are shown in Figure 2a. An initial rapid increase (0–2 h) and a final slight decrease (8–24 h) of WSOC for the three types of aged BB-produced smoke particles can be observed. However, some apparent differences were also identified for the three BB samples from 2 to 8 h. During this period, the $WSOC_{[t]}/WSOC_{[0]}$ values first decreased from 1.05 and 1.07 after 2 h, to 0.99 and 1.04 after 4 h, and then increased to 1.06 and 1.09 after 8 h for the RS and CS samples, respectively. In comparison, the $WSOC_{[t]}/WSOC_{[0]}$ values increased from 1.16 after 2 h to 1.22 after 4 h, and then decreased to 1.18 after 8 h for the PW samples.

According to previous studies, the $O_3$ oxidation of soot particles could produce some new WSOC fractions (Decesari et al., 2002; Li et al., 2015), and $O_3$ may also destroy the C=C bonds within weak- and even non-polar organic compounds to generate newly highly hydrophilic O-containing functional components mainly through oxygen addition (i.e. carboxyl, hydroxyl, carbonyl and ph-OH groups) (Baduel et al., 2011). On the other hand, $O_3$ could lead to C–C bond cleavage within OC fractions to produce many volatile species, such as volatile organic compounds, CO and $CO_2$, resulting in a loss of carbon mass (D'Anna et al., 2009). Due to the complex composition of BB emissions, the formation and decomposition of WSOC simultaneously occurs, and its composition dynamically changes. The initial rapid increase of WSOC content indicates that the formation of WSOC from the oxidation of the soot particles is likely the dominant reaction, and the observable decreasing trends of WSOC content might indicate that the amount being formed is unable to compensate for the amount that decomposes.

[Figure 2]

Changes in the total absorption at 365 nm ($abs_{365}$) were investigated to illustrate the evolution of the chromophores in BB-BrC during the $O_3$ aging process (Figure 2b). The $abs_{365}$ values of BrC for the three BB smoke samples all gradually decreased with an increase of the aging time. As shown in Figure 2b, a 41%–49% decrease of $abs_{365}$ was seen for 24 h of $O_3$ aging, indicating that the BB-BrC underwent significant bleaching. Given that some increase in the WSOC content was observed during the $O_3$ aging process, the decreased $abs_{365}$ may have been driven by a loss of highly absorbent WSOC and/or the formation of non-/weakly absorptive WSOC fractions (Eugene et al., 2016; Lavi et al., 2017; Pillar et al., 2014, 2015; Rincón et al., 2009, 2010; Xia et al., 2018). This was consistent with the previous findings that the bleaching of BB-derived chromophores, such as oxocarboxylic acids and oxy-aromatics, generally resulted in formation of non-light absorbing compounds, such as aliphatic mono- or polycarboxylic acids, containing alcohol -OH, aldehyde -C(H)=O, and ether >O groups (Eugene et al., 2016; Lavi et al., 2017; Pillar et al., 2014, 2015; Rincón et al.,

2009, 2010; Xia et al., 2018). Moreover, the changes of $abs_{365}$ for the three types of BB-BrC also exhibited some differences. The PW-BrC presented higher degrees of bleaching than the CS- and RS- BrC (Figure 2b). This suggests that the bleaching of chromophoric BrC within PW smoke particles during $O_3$ aging are much more likely to occur than for CS and RS smoke particles.

3.2. Variations of the chromophoric characteristics of BrC

In the present study, $SUVA_{254}$, HIX, AAE and $MAE_{365}$, were investigated to characterize the evolution of the aromaticity, humification degree, and light-absorbing properties of BB-BrC during $O_3$ aging. As shown in Figure 3a, the BB-BrC $SUVA_{254}$ values gradually declined, especially during the first hour, implying a significant decrease of aromaticity during $O_3$ aging. This may be attributed to the decomposition of aromatic species and/or the formation of many more hydrophilic organic compounds with weak- or even non-light absorption properties. It agreed well with the observations that the $O_3$ oxidation of BB-derived chromophores (i.e. oxy-aromatics) could lead to cleavage of aromatic bonds into polyfunctional low molecular weight carboxylic acids (Pillar et al., 2014, 2015, 2017). The noticeable HIX increases seen for the three types of BB-BrC indicate that the $O_3$ aging may strongly decompose the protein-like fluorophores, probably phenolic compounds (Chen et al., 2016a), to form polyhydroxylated aromatic species or newly humic-like fluorophores (Pillar et al., 2014, 2015, 2017; Decesari et al., 2002; Li et al., 2013) (Figure 3b). For example, $O_3$ oxidation of phenolic compounds could form polyhydroxylated aromatic compounds with absorption red-shift, which might lead to their HIX values increase (Lavi et al., 2017; Magalhães et al., 2017; Pillar et al., 2015; Rincón et al., 2009, 2010). There seems to be contradictions between the results of the $SUVA_{254}$ and HIX analyses, but together they are reasonable. It is known that $SUVA_{254}$ reflects the average aromaticity of entire WSOC fractions (Fan et al., 2016, 2018), which can be greatly reduced by weakly/non-light absorbing organic compounds formed during $O_3$ aging. However, HIX represents the humification characteristics of fluorophores within aged BrC (Qin et al., 2018), of which the influence of non-chromophores is excluded.

[Figure 3]

The $MAE_{365}$ values of the BB-BrC showed a gradual decline as a function of oxidation time (Figure 3c), giving rise to the $MAE_{365}$ values of the 24-h aged BB-BrC being reduced by a factor of 1.6–2.2. This suggests that the absorption efficiency of BrC was weakened by the $O_3$ aging, which greatly resembles the bleaching behaviors of BB-derived BrC induced by photochemical oxidation and even complex real-world atmospheric processes (Chen et al., 2018; Kumar et al., 2018). For example, OH radical oxidation of wood-burning emissions led to the mass absorption cross-section at 370 nm of BrC declining by up to 2.3 times (Kumar et al., 2018). The BB-BrC AAE values were observed to change during $O_3$ aging as well (Figure 3d). The RS- and CS- BrC AAE values generally increased as the $O_3$ oxidation proceeded,

suggesting that the light absorption of aged BrC exhibited a stronger wavelength dependence than fresh BrC. This result is consistent with the behaviors of BB-derived BrC under photo-chemically aged processes reported in previous studies (Rincón et al., 2010; Saleh et al., 2013). It is noted that the PW-BrC AAE values increased during the first 2 h, but gradually declined during the next oxidation period. This behavior might be ascribed to the chromophores in crop straw and wood burning produced BrC fractions having some differences in their respective compositions, which can be seen in the UV-vis (Figure S3) and SF spectra (Figure S4) and those presented in previous studies (Fan et al., 2016, 2018).

## 3.3. EEM and PARAFAC analysis

As shown in Figures S5–S7, all the initial EEMs of the BB-BrC display two apparent peaks, which are located in the range of $\lambda_{ex}/\lambda_{em}$ = 230–240 nm/365–390 nm and 260–270 nm/360 nm. These two peaks are usually assigned to tyrosine-like compounds and tryptophan-like compounds, respectively, which are widely found in water soluble organic matters (WSOM) and humic-like substances (HULIS) in BB aerosols (Fan et al., 2016; Huo et al., 2018), atmospheric aerosols and rainwater (Qin et al., 2018; Santos et al., 2012). During the $O_3$ aging process, these two fluorescence peaks in the BB-BrC EEMs gradually shifted to longer wavelength regions as the oxidation time increased. After 24 h of $O_3$ aging, the above two peaks had almost disappeared and were replaced by the appearance of two apparent peaks in the range of $\lambda_{ex}/\lambda_{em}$ =240–250 nm/400–425 nm and 280–290/390–400 nm for the three BB-BrC samples (Figures S5–S7). These two peaks are usually assigned to humic-like fluorophores (Qin et al., 2018; Santos et al., 2009, 2012), suggesting the formation of new fluorophores in the aged BB-BrC. It is obvious that the resulting spectral characteristics are very similar to those of HULIS and WSOM in atmospheric aerosols and rainwater (Qin et al., 2018; Santos et al., 2009, 2012). Therefore, it can be concluded that $O_3$ aging leads to a significant transformation of BB-BrC, from dominant protein-like fluorophores within fresh BB-BrC to dominant humic-like fluorophores within aged BB-BrC.

Quantified variations of the independent fluorophores can be revealed with an EEM-PARAFAC analysis. On the basis of a comparison of the PARAFAC-derived components for atmospheric the WSOM and HULIS (Chen et al., 2016a, b), BB-produced WSOM and HULIS (Huo et al., 2018), biochar- and compost-produced dissolved organic matters (DOM) (Jamieson et al., 2014; Huang et al., 2018), four fluorescent components were identified for the fresh and $O_3$ aged BB-BrC extracts. Their profiles and the corresponding assignments are shown in Figure 4a and Table S1, respectively. In general, these four individual fluorophores can be attributed to long-wavelength humic-like chromophores (HULIS-1, C1), short-wavelength humic-like chromophores that are less oxygenated (HULIS-2, C2) and highly oxygenated (HULIS-3, C4) and protein-like or phenol-like organic matter (PLOM, C3), respectively (Chen et al., 2016a, b; Gao et al., 2017; Huang et al., 2018; Huo et al., 2018; Jamieson et al., 2014). It is noted that HULIS-1, -2, and -3 are commonly

found in the EEMs of HULIS and WSOM in atmospheric aerosols and rainwater (Chen et al., 2016b; Matos et al., 2015), which are expected to consist of fluorophores with strong aromaticity and large molecular sizes. PLOM are mainly comprised of the fluorophores owning similar position of fluorescence peaks to proteins, which generally include nitrogen-containing compounds (e.g. atmospheric amines and amides) and also non-nitrogen-containing species (e.g., phenol- and naphthalene-like substances) (Chen et al., 2016a; Fan et al., 2016; Huo et al., 2018). It is noted that the identified PLOM spectra of the $O_3$-aged BrC resemble to those of fresh BrC, suggesting protein-like and/or phenol-like substances are important fluorophore within both the fresh and aged BB-produced BrC.

[Figure 4]

The quantitative analysis of the PARAFAC-derived components of the BB-BrC upon $O_3$ aging are displayed in Figure S8. It can be seen that the $O_3$ aging caused significant quenching for components C1–C3 for all the BB-BrC samples, in which the quenching of C3 is the most dominant with a degree range of 71% to 94%. These findings suggest that the PLOM are susceptible to $O_3$ aging. Noteworthy, the component C4 exhibits a gradually increase with $O_3$ aging for the RS- and CS- BrC. This suggests that HULIS-3 is generated with an accompaniment of decomposition of PLOM, HULIS-1 and HULIS-2 for crop straw burning BrC. In comparison, HULIS-3 (C4) presented an obvious decrease for the PW-BrC, indicating that all the fluorophores in the samples were susceptible to $O_3$ aging. The variations in the proportional distribution of the independent components during $O_3$ aging can be observed in Figure 4b. In terms of the proportion of HULIS-1 (C1), the BrC samples during $O_3$ aging presented relative higher values than fresh ones, but there were no significant differences among the aged samples. The proportion of PLOM (C3) within the BB-BrC gradually declined during $O_3$ aging, where the total decrease was from 37% to 16% for RS, 45% to 16% for CS and 51% to 13% for PW. In comparison, the proportion of HULIS-3 (C4) gradually increased from 0% to 16% for RS, 11% to 31% for CS and 24% to 51% for PW. Taken together, the distributions of the humic-like fluorophores (HULIS-1, -2, -3) accounted for 84%–87% of the aged BB-BrC, which is much higher than the range 49–63% observed for the initial fresh samples. It had been revealed that the $O_3$ oxidation of oxy-aromatics could produce polyhydoxylated aromatics with light absorption toward to longer wavelengths (Lavi et al., 2017; Magalhães et al., 2017; Pillar et al., 2015). These oligomers might have similar fluorescence peak to those of the humic-like fluorophores, so that the humic-like fluorophores identified herein might also be ascribed to the polyhydroxylated aromatics. These results indicate that humic-like fluorophores or possible polyhydroxylated aromatics in BB-BrC are gradually enriched during $O_3$ aging. This is in good agreement with the findings from the enhanced humification of BB-BrC with the increased HIX values after $O_3$ aging. Importantly, the resulting humic-like fluorophore distribution is similar to the 84%–85% range reported in previous field WSOM measurements (Chen et al., 2016b; Matos et al., 2015), which thus implies that the fluorophore distributions for aged BB-BrC are similar to relevant atmospheric

distributions. In addition, it is obvious that the quenching behavior of each fluorophore is highly different among the three types of BB-BrC during $O_3$ aging. For example, smaller distributions of HULIS-1 and larger distributions of HULIS-3 in the aged PW smoke BrC were found relative to those in the aged crop straw smoke BrC (Figure 4b). This implies that PW-BrC contains less high-molecular-weight HULIS and more low-molecular-weight HULIS with highly oxygenated.

### 3.4. 2D-COS combined with fluorescence and FTIR spectra

Two-dimensional correlation spectroscopy was applied to the series of SF and FTIR spectra to tentatively interpret the $O_3$ aging mechanism of BB-BrC. As shown in Figure S9, the major autopeaks centered at 267nm/289 nm, 284 nm and 276 nm are present in the RS-, CS- and PW- BrC synchronous maps, respectively. These peaks are all within the protein-like fluorescent region, suggesting that the protein-like substances in the samples are very susceptible to $O_3$ aging. These results agree well with the significantly reduced PLOM found from the PARAFAC-derived fluorophore analysis. The asynchronous maps further revealed the sequential changes of the fluorophores within the BrC, as illustrated in Figure S10. According to Noda's rule and the signs of the cross-peaks given in Table S2, the order of fluorophores changes are ("$>$" means prior to): 267 nm > 289 nm > 333 nm for the RS-BrC, 263 nm > 284 nm > 340 nm for the CS-BrC, and 309 nm > 276 nm > 358 nm for the PW-BrC. The fluorescence regions in the wavelength ranges of 250 to 300 nm and 300 to 380 nm are usually attributed to protein-like and fulvic-like fluorophores, respectively (Chen et al., 2015; Jin et al., 2018; Pantelaki et al., 2018). It is obvious that the RS- and CS- BrC exhibited the same susceptible fluorophore changes upon $O_3$ aging, showing that the oxidation of the protein-like fraction (263–289 nm) occurred earlier than that of the fulvic-like fraction (333–340 nm). For the PW-BrC, the $O_3$ aging of the fulvic-like fluorophores with shorter wavelengths (309 nm) took place before the protein-like fluorophores (276 nm), which was followed by the fulvic-like fluorophores with longer wavelengths (358 nm). The results also imply that the reaction sites in the PW-BrC under $O_3$ aging were different from those in crop straw burning BrC. Taken together, it can be concluded that the protein-like substances were susceptible fluorophores, which were primarily oxidized before the longer-wavelength fulvic-like substances. This, however, is in contrast with the results from the dark OH radical oxidation of BB-BrC reported in our recent study (Fan et al., 2019), suggesting that the fluorophores within the BrC oxidized by OH in the aqueous phase and those by $O_3$ in the particle-phase are different.

The main changes of the FTIR spectra of the BB-BrC upon $O_3$ aging are displayed in the range 1800–1000 cm$^{-1}$ (Figure 5). Detailed information regarding any variations in the intensity and sequence of the functional groups upon $O_3$ aging can be revealed with 2D-FTIR-COS, where synchronous and asynchronous maps were obtained, as shown in Figure 5. The position and signs of the auto-peaks and cross-peaks are summarized in Table

S3. In general, the synchronous maps exhibit one auto-peak at 1725 cm$^{-1}$ for RS-BrC, two auto-peaks at 1726 and 1639 cm$^{-1}$ for CS-BrC, and two autopeaks at 1725 and 1630 cm$^{-1}$ for PW-BrC. These findings suggest that carboxylic C=O (1725, 1726 cm$^{-1}$) and aromatic C=C and C=O groups (1630, 1639 cm$^{-1}$) (Yan et al., 2013;Fan et al., 2016;Fan et al., 2013;Chen et al., 2015;Zhao et al., 2016) are very susceptible to O$_3$ aging. This result can be confirmed by the evolution behaviors of O$_3$ oxidation of BB-derived phenolic compounds (Pillar et al., 2014, 2015, 2017). For example, O$_3$ primarily attacked the C=C bond in such oxy-aromatics to yield polyfunctional low molecular weight carboxylic acids containing C=O groups (Pillar et al., 2014, 2015, 2017). In addition, the cross-peaks generally showed a positive correlation between the series of bands at 1725 (1726), 1639, 1630, 1400, 1318 and 1224 (1211) cm$^{-1}$. This implies that the stretching of carboxylic C=O (1725, 1726 cm$^{-1}$), phenol-OH (1400, 1224, 1211 cm$^{-1}$), and aromatic C=C, ketone and amide C=O groups (1630, 1639 cm$^{-1}$) (Yan et al., 2013;Fan et al., 2016;Fan et al., 2013;Chen et al., 2015;Zhao et al., 2016;Wang et al., 2017) were co-transformed upon O$_3$ aging. It is also in good agreement with the ozonolysis pathway for BB-derived oxy-aromatics (Pillar et al., 2014, 2015, 2017). For example, the O$_3$ oxidation of oxy-aromatics can cause the cleavage of the aromatic bond (C=C) to generate polyfunctional low molecular weight carboxylic acids (C=O), and also formation of polyhydroxylated aromatics (phenol-OH). Moreover, the cross-peaks also presented a positive correlation between at another series of bands at 1585 (1580) and 1515 cm$^{-1}$ also, suggesting that the vibration of aromatic COO$^-$ (1580, 1585 cm$^{-1}$) and lignin skeletal C=C (1515 cm$^{-1}$) (Fan et al., 2016;Yan et al., 2013;Wang et al., 2017) were also changed in the same way. However, there is a negative correlation between these two series of bands, implying that the spectral changes proceeded in reverse. Given the changes seen in the FTIR spectra (Figures 5a1–c1), we can concluded that the former groups (i.e. carboxylic C=O, phenol-OH, protein C=C and C=O, and aromatic ketone C=O groups) are more like to be generated, while the lignin structures (i.e. aromatic COO$^-$ and lignin skeletal C=C) are tended to be decomposed. Actually, this is consistent with observations of more oxygen-containing functional groups (i.e. ketones, aldehydes, and anhydrides) that formed in soot particles after O$_3$ oxidation reported in many previous studies (He et al., 2017; Li et al., 2013, 2015). For example, He et al. (2017) pointed out that O$_3$-aged soot particles presented an increase in the intensities of the bands at 1715, 1630 and 1055 cm$^{-1}$ in their FTIR spectra, suggesting the formation of ketone C=O and C-O groups. The increase of oxygen-containing functional groups might lead to the solubility of OC, which is in good agreement with the former observations of much more WSOC being generated during the O$_3$ aging of BB-smoke particles.

[Figure 5]

The asynchronous maps displayed more information regarding changes of the functional groups. The major cross-peaks and their signs are listed in Table S3. According to Noda's rule, the sequence of bands upon O$_3$ aging are (">" means prior to) 1224 > 1725 > 1580 > 1515 cm$^{-1}$ for the RS-BrC, 1400, 1211 > 1726 > 1639 > 1515 cm$^{-1}$ for the CS-BrC, and 1630 >

1725 > 1585 > 1515, 1318 cm$^{-1}$ for the PW-BrC. These indicate that different functional groups within the BrC formed or changed during O$_3$ aging. Nevertheless, the similar structural change sequence of the BB-BrC functional groups upon O$_3$ aging could be ascertained. The changes follow the order of aromatic OH (1400, 1224, 1211 cm$^{-1}$) > conjugated C=O groups (1725, 1726 cm$^{-1}$) and aromatic C=O (1630, 1639 cm$^{-1}$) > aromatic COO$^-$ (1585 and 1580 cm$^{-1}$) > lignin-derived C-C, C-H and C-O groups (1515 and 1318 cm$^{-1}$). Actually, the conjugated C=O formation were observed to be occurred before aromatic C=O formation for the CS-BrC (1726>1639 cm$^{-1}$) during O$_3$ aging, while the reverse sequence were observed for the PW-BrC (1630>1725 cm$^{-1}$). These results suggest that the aromatic C=O functional groups involved with O$_3$ aging are different for these two types of BB-BrC. As revealed by our previous study, the pyrolysis of CS- and PW- HULIS both dominantly generated lignin derivatives, but amounts of diterpenoid derivatives were also formed by pyrolysis of PW-HULIS (Fan et al., 2016). Therefore, it can be speculated that the 1639 cm$^{-1}$ feature can more likely be attributed to the stretching of aromatic C=O, linked to the lignin derivative of the CS-BrC, while the 1630 cm$^{-1}$ feature can more likely be ascribed to the stretching of aromatic C=C and C=O, linked to the diterpenoid derivatives of the PW-BrC.

**4. Conclusions and atmospheric implications**

In this study, the O$_3$ aging of BB smoke particles from the burning of RS, CS and PW were conducted to investigate the evolutionary behavior of BrC. The results showed that the WSOC content of BB-BrC quickly increased during the initial O$_3$ aging period but slowly decreased during the final O$_3$ aging period. Simultaneously, the O$_3$ aging led to a gradual reduction of absorption and fluorescence for all the BB-BrC samples, suggesting the significant degradation of chromophores and fluorophores. Moreover, a decrease of aromaticity, light-absorbing ability of the chromophores, and an increase of humification of the fluorophores within the BB-BrC were also observed during the aging process. Therefore, it might be concluded that both the degradation and formation of water-soluble OC fractions occur during O$_3$ aging.

The EEM-PARAFAC analysis revealed that HULIS-1, HULIS-2, and PLOM within the BB samples were significantly degraded, but newly formed HULIS-3 was obviously observed for two of the RS and CS samples. In addition, the relative distribution of PLOM gradually decreased, and that of HULIS-3 was observed to gradually increase for all the BB-BrC samples. This indicates that the aged BB-BrC contains more humic-like substances with highly oxygenated chromophores, but fewer protein-like substances. These results also confirm that the fluorophore composition of BrC is altered by the O$_3$ aging of the BB particles, and more humic-like substances (HULIS-1,2,3) are abundant in aged BB-BrC. It is noted that the fluorescent components of the aged BB-BrC are quite similar to those of atmospheric BrC.

Two-dimensional correlation spectroscopy of the SF and FTIR spectra revealed valuable information regarding the reactive positions and sequences within BB-BrC for $O_3$ aging. Although the different chemical compositions of BB-BrC resulted in different evolutionary behaviors, some similar $O_3$ aging mechanisms could be tentatively identified. The results from 2D-SF-COS analysis suggest that the protein-like fraction (263–289 nm) is susceptible to $O_3$ aging before the fulvic-like fraction (333–340 nm) within the RS- and CS- BrC. The PW-BrC presented susceptible fluorophores in the order of fulvic-like fluorophores with shorter wavelengths (309 nm) > protein-like fluorophores (276 nm) > fulvic-like fluorophores with longer wavelengths (358 nm). The 2D-FTIR-COS analysis revealed the sequence of aromatic OH (1400, 1224, 1211 cm$^{-1}$) > conjugated C=O groups (1725 cm$^{-1}$) and aromatic C=O (1630, 1639 cm$^{-1}$) > aromatic COO$^-$ (1585 and 1580 cm$^{-1}$) > lignin-derived C-C, C-H and C-O groups (1515 and 1318 cm$^{-1}$) for all the BB-BrC. This implies that the formation of phenols occurred before the formation of carboxylic C=O and aromatic C=O through O addition, which were then accompanied by the decomposition of lignin derivatives for all the BB-BrC during $O_3$ aging.

Given the high concentrations of $O_3$ and large amounts of BB emission in the atmosphere, $O_3$ aging processes often occur in atmospheric environment. The results obtained in this study provide new insights into the evolutionary behavior of the chromophoric and fluorescent properties of BB-BrC during $O_3$ aging, which have important implications in terms of the heterogeneous oxidation of BB-BrC. Furthermore, previous modeling studies related to BB-BrC only focused on parametrizing the optical properties of fresh BB-BrC, and the consideration of effects related to atmospheric processes on BB-BrC are limited. The present study had revealed that a considerable bleaching rather than competitive formation of chromophoric BrC occurred during $O_3$ aging of BB smoke samples, indicating that the relevant oxidation chemistry taking place in the atmospheric gas phase will weaken the light absorption properties of BB-derived BrC. As a result, the radiative forcing potential of BB-derived BrC is likely overestimated. These findings are of great importance for improving the accuracy of climate models as well as source apportionment models that consider the optical properties of BrC. However, some questions are still remained and more studies should be conducted in the future: (1) the $O_3$ oxidation mechanism of BB-BrC under different conditions (e.g., $O_3$ concentration, RH, temperature), especially under real atmospheric environment; (2) the $O_3$ oxidation mechanism of BrC derived from other sources such as fossil fuel combustion, secondary chemical formation, etc.

*Data availability*. All data needed to evaluate the conclusions in the paper are present in the paper and Supplement. Additional data related to this paper may be requested from the authors.

*Author contributions*. JS, PP and XF conceived of the experiment. TC and XY built the ozone

aging reactor and operate ozone aging of biomass burning fine particles. TC, XY, YW, WJ performed TOC, UV-vis, 3DEEM and FTIR measurements. XF and JS performed 2DCOS and PARAFAC analysis. XF interpreted all data with assistance of JS. XX, FL, YW and YX provided useful comments on the paper. XF wrote the paper with assistance of JS and PP.

*Competing interests.* The authors have no competing interests to declare.

*Acknowledgements.* This study was supported by the Natural Science Foundation of China (Grants numbers 41705107, 41673117); the Anhui Science and Technology Major Project (Grant number 16030701102); the Anhui Provincial Natural Science Foundation (Grants numbers 1808085MB49).

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

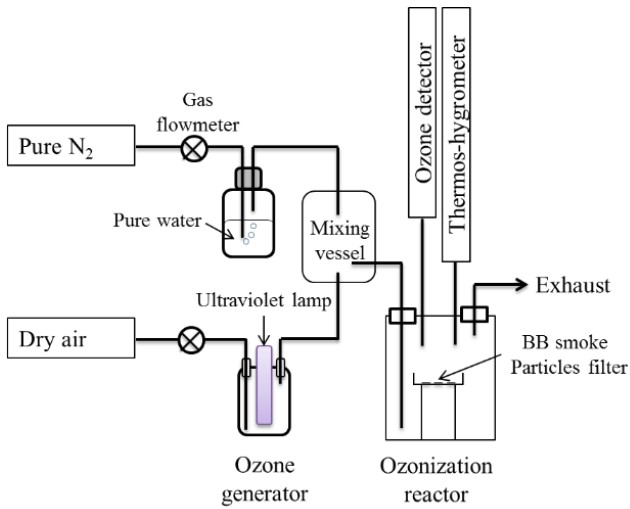

Figure 1. The experimental apparatus.

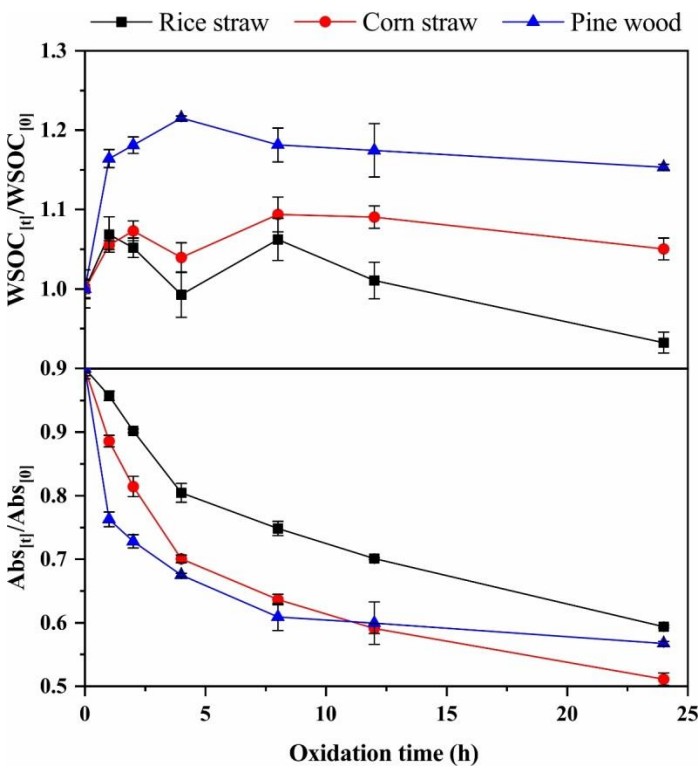

Figure 2. Dynamic variations of (a) WSOC content, (b) light absorption at a wavelength of 365 nm by the BB-BrC during the ozone aging process.

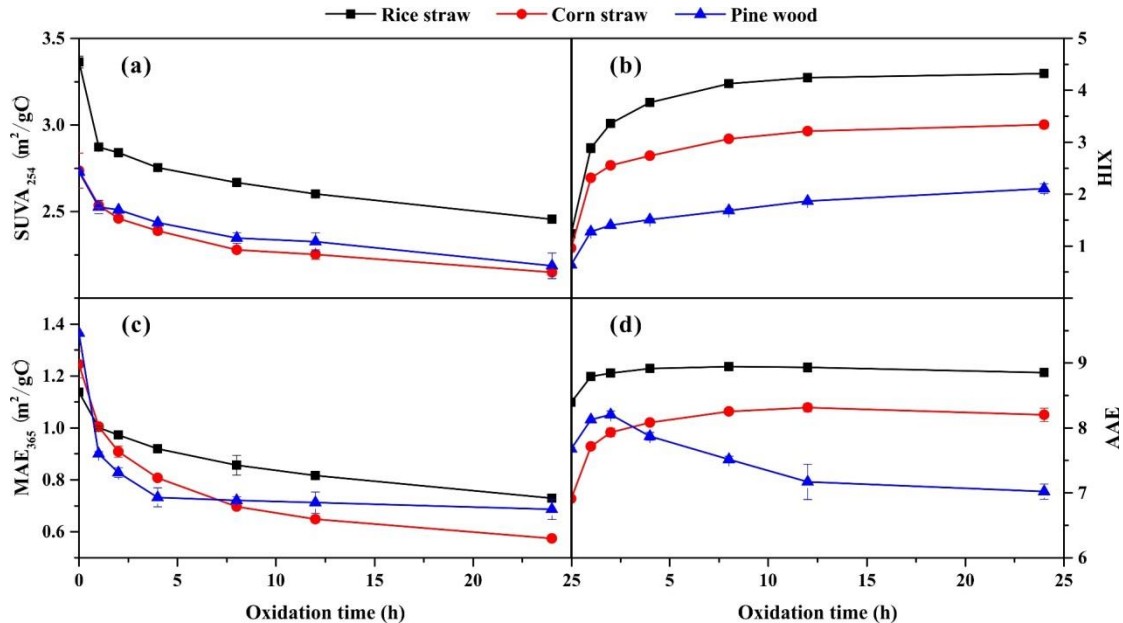

Figure 3. Dynamic variations of (a) SUVA$_{254}$, (b) HIX, (c) MAE$_{365}$, and (d) AAE of RS-, CS- and PW- smoke water-soluble BrC during the ozone aging process.

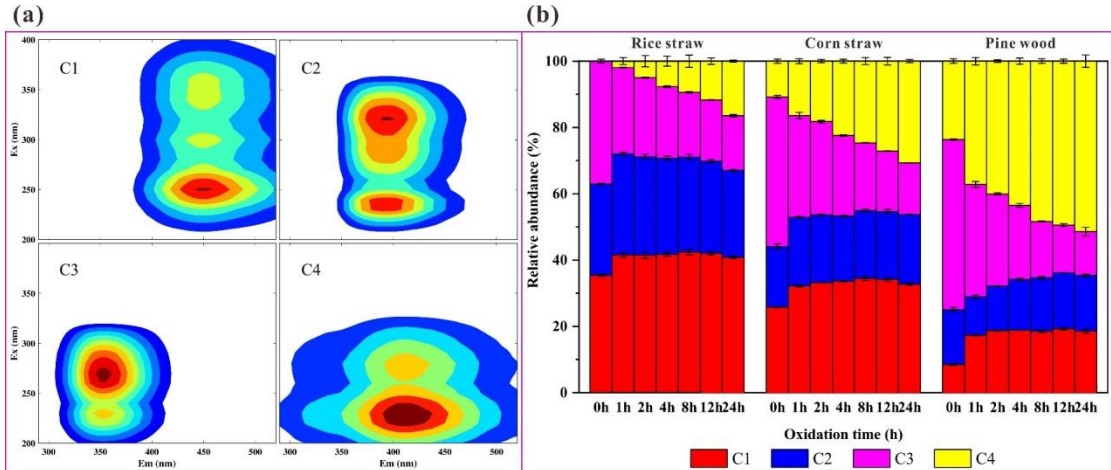

Figure 4. (a) Three fluorescent components of the BB-BrC identified by the EEM-PARAFAC model. (b) Variations in the abundances of the individual fluorescent components (C1–C4) within the water-soluble BrC in RS, CS and PW smoke particles during the ozone aging process. The fluorescent spectra of components C1–C4 are listed in the upper layer, and the corresponding loading positions and identification are shown in Table S1 .

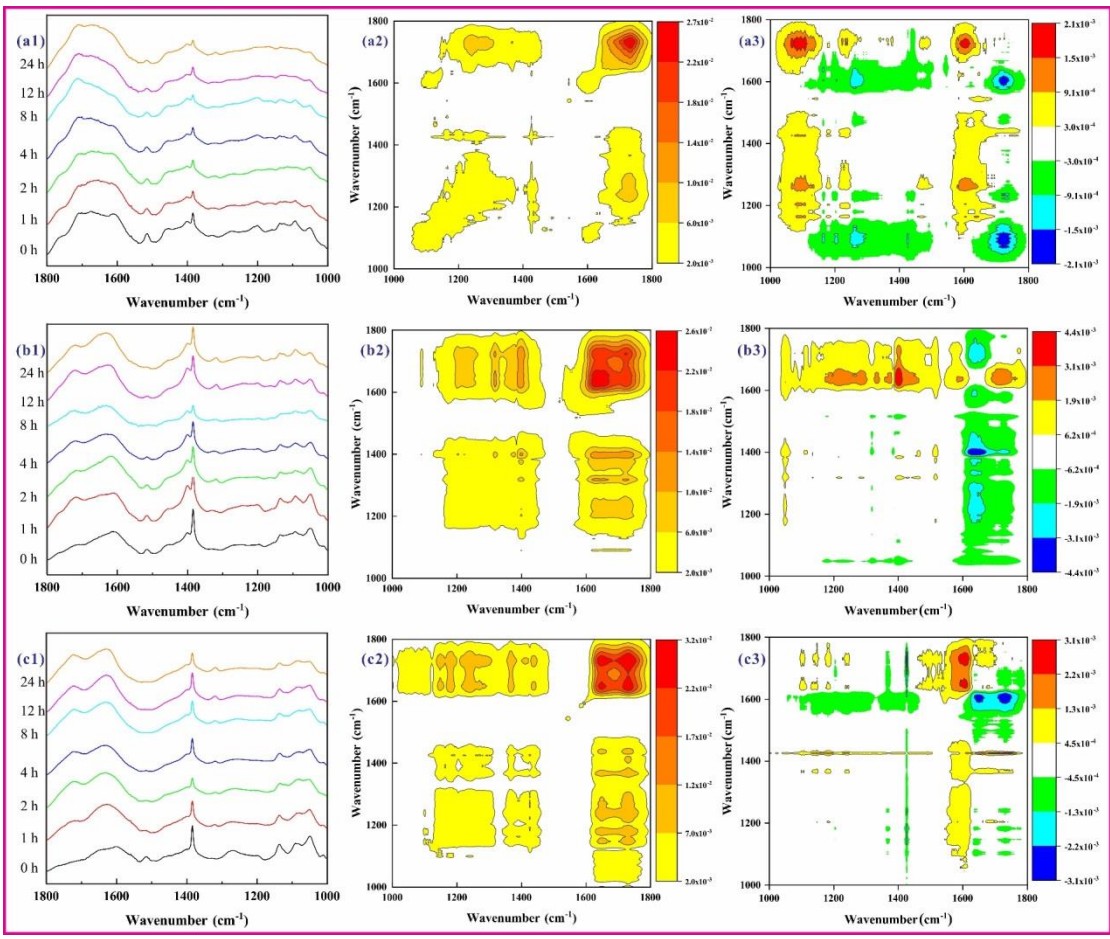

Figure 5. FTIR spectra, and the corresponding synchronous and asynchronous 2D FTIR COS maps for the RS- (a1-3), CS- (b1-3) and PW- (c1-3) BrC during the ozone aging process.