# Peer review of "The evolutionary behavior of chromophoric brown carbon during ozone aging of fine particles from biomass burning"

_Atmospheric Chemistry and Physics, 2019_

## Referee Comment (RC1) · Anonymous Referee #1 · 17 Jan 2020

This manuscript, titled "The evolutionary behavior of chromophoric brown carbon during ozone aging of tine particles from biomass burning", communicates an in-depth study of brown carbon particulate matter from three fuels, and the consequences of ozone aging. I am impressed by the depth of the study and the thoughtful discussion in the results section. I believe this work is well-suited for publication in Atmospheric Chemistry and Physics, however, I have three concerns that should be addressed before the manuscript be accepted for publication.

Major criticisms and questions

1. The ozone aging experiments are described as taking place for a set of "designed

exposure times" (page 5, line 20). In the context of atmospheric aerosol aging, it is desirable (if not necessary) to equate reactor times to equivalent atmospheric aging. This is essential to drawing conclusions between laboratory results and in situ, real world observations. I am concerned that exposing the filters to 70 ppm of ozone is not strictly applicable to the real-world atmosphere, where concentrations exceeding 8 ppm are rare, and only found in the upper stratosphere. Some analogy between time spent in the reactor to time spent in the atmosphere would greatly strengthen this manuscript. Can this information be provided, along with the methodology used to derive it? The specific methodology may be relegated to the supporting information.

2. Quartz filters are known to collect some fraction of the gas phase (e.g., Parshint-sev et al. 2011). Depending on how quickly the ozone aging was performed after filter gathering was performed (and the storage and handling methods of the filters), gas-phase artifacts (SVOCs and IVOCs especially) may be interacting with the reactor environment in ways that aren't fully explored in the manuscript. The authors should discuss this and other sources of filter artifacts.

3. On page 13, line 20, the authors state "The present study has confirmed that the bleaching of chromophoric BB-BrC dominantly occurs during O3 aging..." While the authors have presented strong evidence that O3 aging can certainly bleach BB-BrC, they present no evidence that it is the dominant mechanism for bleaching. In fact, on page 8, they present contradictory evidence from Kumar et al. (2018) who showed that the MAE365 values decreased by up to 2.3 times under OH radical aging, whereas in this study they showed a maximum decrease of 2.2. The authors should address this discrepancy, and provide a thorough meta-analysis of bleaching results from literature investigating different pathways and oxidants.

Minor technical corrections and clarifying questions Page 5 Line 3 – I suggest re-writing the final sentence to "BB particles were obtained from each of the three fuels." Line 10 – I am unfamiliar with the term "glass garden". Please explain what this is, and its specific use in the ozone aging experiments. Line 23 – replace "designed" with

"designated".

Page 6 Line 19 – replace "series" with "type". Line 30 – remove "It is obvious that".

Page 7 Line 13 – remove "It is obvious that".

Page 11 Line 2 – Change "Detail" to "Detailed".

References Parshintsev, J., Ruiz-Jimenez, J., Petäjä, T. et al. Comparison of quartz and Teflon filters for simultaneous collection of size-separated ultrafine aerosol particles and gas-phase zero samples. Anal Bioanal Chem 400, 3527–3535 (2011) doi:10.1007/s00216-011-5041-0

---

## Referee Comment (RC2) · Anonymous Referee #2 · 29 Jan 2020

General Comments:

The manuscript presents a laboratory study characterizing the evolution of chromophores and fluorophores of material collected in quartz filters during the combustion of rice straw (RS), corn straw (CS) and pine wood (PW) exposed to O3. The work interprets that O3 causes a reduction in light absorption and fluorescence of biomass burning (BB) brown carbon (BrC) material (captured in the filters), which is associated to a loss of aromaticity with a drop in average molecular weight. The manuscript needs some clarification and improvement in the writing and should considerably improve by connecting the findings with literature that is missing. Specific comments for a major

revision recommended are provided below.

Specific Comments:

1) In the abstract (in page 1), there are some inaccuracies and confusing terms. For example, in l. 2, how much of "little is known..." really? There is some literature missing that has not been considered. In l. 4, what is the meaning of "the transformation of levels..."? In l. 11, what are the protein-like components of BB? Is there really much proteins or what do the authors want to explain in the manuscript? Finally, a strong statement is used in l. 27-30 but no connections has been directly provided in the manuscript about how to use the data to parametrize the optical properties of BrC in climate models.

2) Following p. 3 l. 11-29: When explaining that BB-BrC have been studied in the laboratory, the manuscript should connect to the work in the following seven papers by adding a short paragraph summarizing their findings: a) E.A. Pillar et al., Environ. Sci. Technol., 2017, 51, 4951-4959. b) J. Sun et al., Sun, Chem. Eng. J., 2019, 358, 456-466. c) E.A. Pillar et al., Environ. Sci. Technol., 2014, 48, 14352-14360. d) E.A. Pillar et al., J. Phys. Chem. A, 2015, 119, 10349-10359. e) A. Lavi, ACS Earth & Space Chem., 2017, 1 (10) , 637-646. f) A.C.O. Magalhães, ACS Earth & Space Chem., 2017, 1, 353-360. g) Pillar-Little, Environments (2018), 5(9), 104.

It would be fundamentally important to connect the findings of the aromatic structures in the above seven papers with the material in the revision at multiple points of the manuscript such as p. 7 l. 29 and l. 34; p. 9 l. 10, l. 18, and l. 25; p. 10 l. 1; p. 11 l. 10 and l. 15.

3) p. 4 l. 38: What is the metal for the metal mesh holding the biomass materials? What is the temperature of the combustor through operation? Is it constant or variable? Indicate in the revised manuscript.

4) p. 5 l. 1: What is the consideration for the gases adsorbed into the quartz filters?
L. 10: glass garden? Do you mean a terrarium? L. 13: Why is the ozone so high (70 ppm) and how is it environmentally relevant? Why is the relative humidity fixed at 40%? Explain in the revised manuscript.

5) p. 6 l. 2-3: Despite mentioning that more details are in the SI, provide literature references for the concepts of SUVA254, AAE, MAE365, and HIX.

6) p. 6 l. 24: Provide software version and company name that sell it.

7) p. 7 l. 37: Despite what is explained, it appears that in this work there is no soot. How is this point connected to the results?

8) p. 7 l. 12-24; p. 8 l. 12-13; and p. 13 l. 21: Changes in absorption with wavelength for chromophores and/or fluorophores, and the concept of photobleaching, have been investigated in the following seminal papers that introduced the ideas and should be discussed here with the addition of a few relevant statements: a) A.G. Rincon at al., J. Phys. Chem. A, 2009, 113, 10512-10520. b) A.G. Rincon at al., J. Phys. Chem. Lett., 2010, 1, 368-373. c) A.J. Eugene et al., J. Phys. Chem. A, 2016, 120, 3817-3826 d) S.-S. Xia et al., J. Phys. Chem. A, 2018, 122, 6457-6466.

9) p. 7 l. 34: What is the meaning of polycondensation? Clarify.

10) p. 30 l. 30: "...is in good..."

---

## Author Comment (AC1) · 8 Mar 2020

This manuscript, titled "The evolutionary behavior of chromophoric brown carbon during ozone aging of fine particles from biomass burning", communicates an in-depth study of brown carbon particulate matter from three fuels, and the consequences of ozone aging. I am impressed by the depth of the study and the thoughtful discussion in the results section. I believe this work is well-suited for publication in Atmospheric Chemistry and Physics, however, I have three concerns that should be addressed before the manuscript be accepted for publication.

Re: We thank the referee for the positive comments and constructive suggestions, we have revised the manuscript according to the comments point by point.

Major criticisms and questions

1. The ozone aging experiments are described as taking place for a set of "designed exposure times" (page 5, line 20). In the context of atmospheric aerosol aging, it is desirable (if not necessary) to equate reactor times to equivalent atmospheric aging. This is essential to drawing conclusions between laboratory results and in situ, real world observations. I am concerned that exposing the filters to 70 ppm of ozone is not

strictly applicable to the real-world atmosphere, where concentrations exceeding 8 ppm are rare, and only found in the upper stratosphere. Some analogy between time spent in the reactor to time spent in the atmosphere would greatly strengthen this manuscript. Can this information be provided, along with the methodology used to derive it? The specific methodology may be relegated to the supporting information.

Re: Thanks. We agreed with your comments "it is desirable to equate reactor conditions and times to equivalent atmospheric aging, which is essential to drawing conclusions between laboratory results and in situ, real world observations". In fact, many $O_3$ oxidation simulation experiments have been conducted to investigate the aging of carbonaceous compounds under different $O_3$ concentration (20 ppb – 12,200 ppm) (Baduel et al., 2011; D'Anna et al., 2009; Pillar et al., 2014, 2017). For example, low $O_3$ concentrations (20 ppb - 6 ppm) had been used for oxidizing the thin films of humic matter (Baduel et al., 2011; D'Anna et al., 2009) and oxy-aromatics (i.e. catechol and its substituted ones) (Pillar et al., 2014, 2017), in which the changes of the uptake coefficient of $O_3$ and the early aging mechanism occurred in air-particle interface had been explored. In addition, to explore the changes of physicochemical properties of particulate samples from combustion process, or to investigate the optical properties of newly formed light absorbing organic compounds during $O_3$ aging process, a relatively high $O_3$ concentrations (20 ppm-12,200 ppm) were commonly used in the simulation experiments (Li et al., 2013, 2015; Decesari et al., 2002; Gallimore et al., 2011; Pillar et al., 2015; Zhu et al., 2019). Importantly, some studies have revealed that the oxidation mechanism at higher $O_3$ concentration is similar to that done at much lower $O_3$ concentration (Pillar et al., 2015; Gallimore et al., 2011). In this study, the main objective is to investigate the evolutionary behavior of chromophoric BrC compounds during $O_3$ aging of BB smoke samples, thus the aging simulation experiment was conducted at a relative higher $O_3$ concentration (70 ppm). The detailed explanation for high $O_3$ concentration used in this study have been provided in "S1. Ozone aging reactor and operation" in the revised supporting information (SI). (see Page 1, lines 13-34 in revised SI)

Moreover, we also agreed with the comments that some analogy between time spent in the reactor to time spent in the atmosphere would greatly strengthen this manuscript. In this study, the $O_3$ exposure amounts for 1 h in reactor were ~$1.7\times10^{15}$ molec $cm^{-3}$ h. For a highly $O_3$ polluted (~120 ppb) area (Chen et al., 2020), 24 h-average atmospheric $O_3$ exposure amount were ~$7.1\times10^{13}$ molec $cm^{-3}$ h. In this case, the oxidation for 1 h in our reactor was approximately equivalent to 260 d of oxidation at polluted atmosphere. However, the smoke samples of this study were highly condensed and coagulated on the filter, so the exposure area of particles were greatly reduced. As a result, the equivalent day for $O_3$ oxidation should be highly shortened. Moreover, our results demonstrated many similar oxidation behaviors of organic chromophores to those of atmospheric humic substances and BB-derived oxy-aromatics under low $O_3$ concentration (20 ppb- 6 ppm) (Baduel et al., 2011; D'Anna et al., 2009; Pillar et al., 2014, 2017). Therefore, we believed that the evolutionary behaviors of BB-BrC revealed here should be similar to those occurred under atmospheric relevant $O_3$ concentrations during their lifetime in atmosphere. Some descriptions have been added in section S1 in revised supporting information. (See Page 1, line 37 to Page 2, line 12 in revised SI).

References:

[1] Baduel, C.; Monge, M. E.; Voisin, D.; Jaffrezo, J.-L.; George, C.; Haddad, I. E.; Marchand, N.; D'Anna, B., Oxidation of Atmospheric Humic Like Substances by Ozone: A Kinetic and Structural Analysis Approach. Environmental Science & Technology 2011, 45, (12), 5238-5244.

[2] Chen, Z.; Li, R.; Chen, D.; Zhuang, Y.; Gao, B.; Yang, L.; Li, M., Understanding the causal influence of major meteorological factors on ground ozone concentrations across China. Journal of Cleaner Production 2020, 242, 118498.

[3] D'Anna, B.; Jammoul, A.; George, C.; Stemmler, K.; Fahrni, S.; Ammann, M.; Wisthaler, A., Light-induced ozone depletion by humic acid films and submicron aerosol particles. Journal of Geophysical Research 2009, 114, (D12).

[4] Decesari, S.; Facchini, M. C.; Matta, E.; Mircea, M.; Fuzzi, S.; Chughtai, A. R.; Smith, D. M., Water soluble organic compounds formed by oxidation of soot. Atmospheric Environment 2002, 36, (11), 1827-1832.

[5] Gallimore, P. J.; Achakulwisut, P.; Pope, F. D.; Davies, J. F.; Spring, D. R.; Kalberer, M., Importance of relative humidity in the oxidative ageing of organic aerosols: case study of the ozonolysis of maleic acid aerosol. Atmos. Chem. Phys. 2011, 11, (23), 12181-12195.

[6] Li, Q.; Shang, J.; Zhu, T., Physicochemical characteristics and toxic effects of ozone-oxidized black carbon particles. Atmospheric Environment 2013, 81, 68-75.

[7] Li, Q.; Shang, J.; Liu, J.; Xu, W.; Feng, X.; Li, R.; Zhu, T., Physicochemical characteristics, oxidative capacities and cytotoxicities of sulfate-coated, 1,4-NQ-coated and ozone-aged black carbon particles. Atmospheric Research 2015, 153, 535-542.

[8] Pillar, E. A.; Camm, R. C.; Guzman, M. I., Catechol Oxidation by Ozone and Hydroxyl Radicals at the Air–Water Interface. Environmental Science & Technology 2014, 48, (24), 14352-14360.

[9] Pillar, E. A.; Guzman, M. I., Oxidation of Substituted Catechols at the Air-Water Interface: Production of Carboxylic Acids, Quinones, and Polyphenols. Environ Sci Technol 2017, 51, (9), 4951-4959.

[10] Pillar, E. A.; Zhou, R.; Guzman, M. I., Heterogeneous Oxidation of Catechol. The Journal of Physical Chemistry A 2015, 119, (41), 10349-10359.

[11] Zhu, J.; Chen, Y.; Shang, J.; Zhu, T., Effects of air/fuel ratio and ozone aging on physicochemical properties and oxidative potential of soot particles. Chemosphere 2019, 220, 883-891.

2. Quartz filters are known to collect some fraction of the gas phase (e.g., Parshintsev et al. 2011). Depending on how quickly the ozone aging was performed after filter gathering was performed (and the storage and handling methods of the filters), gas-phase artifacts (SVOCs and IVOCs especially) may be interacting with the

reactor environment in ways that aren't fully explored in the manuscript. The authors should discuss this and other sources of filter artifacts.

Re: Thanks. We agreed with your comments that the quartz filter could absorb some gas-phase organic artifacts (i.e. semivolatile and intermediate volatility organic compounds) during BB smoke particles sampling (Geller et al., 2006; Parshintsev et al. 2011). According to previous studies, the average adsorption organic artifacts (organic carbon, OC) amounts on quartz filters are very small (0.48-0.98 μgC/cm$^2$) (Arhami et al., 2006; Subramanian et al., 2004). In the current study, the OC contents on fresh BB smoke filters are in the range of ~250-750 μgC/cm$^2$, which are much higher than the possible artifacts. Therefore, the potential contributions from O$_3$ oxidation of possible artifacts on filters to bulk BB-BrC compounds can be neglected. We have added the relevant descriptions in the section 2.2 in revised manuscript. (see Page 4, lines 23-29)

References:
[1] Arhami, M.; Kuhn, T.; Fine, P. M.; Delfino, R. J.; Sioutas, C., Effects of Sampling Artifacts and Operating Parameters on the Performance of a Semicontinuous Particulate Elemental Carbon/Organic Carbon Monitor. Environmental Science & Technology 2006, 40, (3), 945-954.

[2] Geller, M. D.; Ntziachristos, L.; Mamakos, A.; Samaras, Z.; Schmitz, D. A.; Froines, J. R.; Sioutas, C., Physicochemical and redox characteristics of particulate matter (PM) emitted from gasoline and diesel passenger cars. Atmospheric Environment 2006, 40, (36), 6988-7004.

[3] Parshintsev, J.; Ruiz-Jimenez, J.; Petäjä, T.; Hartonen, K.; Kulmala, M.; Riekkola, M.-L., Comparison of quartz and Teflon filters for simultaneous collection of size-separated ultrafine aerosol particles and gas-phase zero samples. Analytical and Bioanalytical Chemistry 2011, 400, (10), 3527-3535.

[4] Subramanian, R.; Khlystov, A. Y.; Cabada, J. C.; Robinson, A. L., Positive and Negative Artifacts in Particulate Organic Carbon Measurements with Denuded

and Undenuded Sampler Configurations Special Issue of Aerosol Science and Technology on Findings from the Fine Particulate Matter Supersites Program. Aerosol Science and Technology 2004, 38, (sup1), 27-48.

3. On page 13, line 20, the authors state "The present study has confirmed that the bleaching of chromophoric BB-BrC dominantly occurs during O3 aging..." While the authors have presented strong evidence that O3 aging can certainly bleach BB-BrC, they present no evidence that it is the dominant mechanism for bleaching. In fact, on page 8, they present contradictory evidence from Kumar et al. (2018) who showed that the MAE365 values decreased by up to 2.3 times under OH radical aging, whereas in this study they showed a maximum decrease of 2.2. The authors should address this discrepancy, and provide a thorough meta-analysis of bleaching results from literature investigating different pathways and oxidants.

Re: Thanks for the comments. We are sorry for this ambiguous sentence. In this study, the results indicated that the bleaching rather than the formation of BrC was the dominant reaction during $O_3$ aging process of BB smoke samples. Therefore, the $O_3$ aging can certainly bleach the BB-BrC compounds, but it didn't mean that the $O_3$ aging was the dominant mechanism for bleaching of BrC in ambient atmosphere. To avoid the misunderstanding, this sentence had been rewritten in revised manuscript. (see Page 13, lines 22-24)

In addition, we think that the results reported by Kumar et al. (2018) don't contradict our findings. The heterogenous aging of BB emission by OH radical simulated by Kumar et al. (2018) and $O_3$ aging in current study both are important reactions in ambient environment. They both indicate the occurrence of bleaching of BrC in the aging process. It is well known that the aging of BB smoke samples may happened with various activated species such as OH radical, $O_3$, or other oxidants in the complex atmospheric environments. The apparent aging changes of BB smoke samples in atmospheric environment should be resulted from complex atmospheric

processes. In this study, we mainly focused on the evolutionary behavior of chromophoric BrC during $O_3$ aging of BB smoke samples. To better understand the aging behaviors of BrC in ambient atmosphere, we have revised the manuscript based on a thorough meta-analysis of bleaching results from literatures investigating different pathways and oxidants in the revised manuscript. (see Page 2, line 39 to Page 3, line 2; Page 6, line 35-39; Page 7, lines 13-16; Page 7, lines 34-35; Page 11, lines 5-9, 14-18)

Minor technical corrections and clarifying questions
Page 5 Line 3 – I suggest re-writing the final sentence to "BB particles were obtained from each of the three fuels."

Re: Revised.

Line10 – I am unfamiliar with the term "glass garden". Please explain what this is, and its specific use in the ozone aging experiments.

Re: We are sorry for this ambiguous word. In this study, it refers to a glass dish. The filter samples were spread in this glass dish and then exposed in ozone environment. The term "glass garden" had been revised to "glass dish ($\Phi$ = 90 mm)" in the text. (see page 4, line 16)

Line 23 – replace "designed" with "designated".

Re: Revised.

Page 6 Line 19 – replace "series" with "type".

Re: Revised.

Line 30 – remove "It is obvious that".

Re: Revised.

Page 7 Line 13 – remove "It is obvious that".

Re: Revised.

Page 11 Line 2 – Change "Detail" to "Detailed".

Re: Revised.

---

## Author Comment (AC2) · 8 Mar 2020

General Comments:

The manuscript presents a laboratory study characterizing the evolution of chromophores and fluorophores of material collected in quartz filters during the combustion of rice straw (RS), corn straw (CS) and pine wood (PW) exposed to O3. The work interprets that O3 causes a reduction in light absorption and fluorescence of biomass burning (BB) brown carbon (BrC) material (captured in the filters), which is associated to a loss of aromaticity with a drop in average molecular weight. The manuscript needs some clarification and improvement in the writing and should considerably improve by connecting the findings with literature that is missing.

Re: We would like to thank the reviewer again for the constructive and thoughtful suggestions. Detailed responses to the individual specific comment/suggestion are as follows.

Specific comments for a major revision recommended are provided below.

Specific Comments:

1) In the abstract (in page 1), there are some inaccuracies and confusing terms. For example, in l. 2, how much of "little is known. . ." really? There is some literature missing that has not been considered.

Re: Thanks for the comments. We agree with the comment that "little is known. . ." is inaccuracy, because some literatures are unconsidered. This sentence has been rewritten in revised manuscript. (see Page 1, lines 2-4)

Page 1, lines 2-4: "Biomass burning (BB) emits large amounts of brown carbon (BrC), however, the evolutionary behavior of BrC in BB emissions (BB-BrC) resulted from complex atmospheric processes are poorly understood."

In l. 4, what is the meaning of "the transformation of levels. . ."?

Re: The "levels" herein refers to the contents. We have changed "levels" into "contents". (see Page 1, lines 4)

In l. 11, what are the protein-like components of BB? Is there really much proteins or what do the authors want to explain in the manuscript?

Re: In this study, the protein-like components were obtained from the excitation-emission matrix combined with a parallel factor analysis (EEM-PARAFAC). The protocol has been widely used to identify BrC components in BB aerosol and ambient aerosol (Chen et al., 2016a, b; Fan et al., 2019; Huo et al., 2018; Matos et al., 2015). These protein-like components mainly included nitrogen-containing compounds, such as amines and amides, and even oxygen-containing compounds, such as phenol- and naphthalene-like substances (Chen et al., 2016a). Therefore, the protein-like components of BB-BrC identified in this study were mainly comprised of the organic substances with similar position of fluorescence peaks to proteins, rather than the real proteins. For better understanding, we have added some descriptions in revised

References:

[1] Chen, Q.; Ikemori, F.; Mochida, M., Light Absorption and Excitation–Emission Fluorescence of Urban Organic Aerosol Components and Their Relationship to Chemical Structure. Environmental Science & Technology 2016a, 50, (20), 10859-10868.

[2] Chen, Q.; Miyazaki, Y.; Kawamura, K.; Matsumoto, K.; Coburn, S.; Volkamer, R.; Iwamoto, Y.; Kagami, S.; Deng, Y.; Ogawa, S.; Ramasamy, S.; Kato, S.; Ida, A.; Kajii, Y.; Mochida, M., Characterization of Chromophoric Water-Soluble Organic Matter in Urban, Forest, and Marine Aerosols by HR-ToF-AMS Analysis and Excitation–Emission Matrix Spectroscopy. Environmental Science & Technology 2016b, 50, (19), 10351-10360.

[3] Fan, X.; Yu, X.; Wang, Y.; Xiao, X.; Li, F.; Xie, Y.; Wei, S.; Song, J.; Peng, P. a., The aging behaviors of chromophoric biomass burning brown carbon during dark aqueous hydroxyl radical oxidation processes in laboratory studies. Atmospheric Environment 2019, 205, 9-18.

[4] Huo, Y.; Li, M.; Jiang, M.; Qi, W., Light absorption properties of HULIS in primary particulate matter produced by crop straw combustion under different moisture contents and stacking modes. Atmospheric Environment 2018, 191, 490-499.

[5] Matos, J. T. V.; Freire, S. M. S. C.; Duarte, R. M. B. O.; Duarte, A. C., Natural organic matter in urban aerosols: Comparison between water and alkaline soluble components using excitation–emission matrix fluorescence spectroscopy and multiway data analysis. Atmospheric Environment 2015, 102, 1-10.

Finally, a strong statement is used in l. 27-30 but no connections has been directly provided in the manuscript about how to use the data to parametrize the optical properties of BrC in climate models.

Re: Thanks. We agree with your comment that no connections have been directly

provided in the manuscript about how to use the data to parametrize the optical properties of BrC in climate models, thus the strong statement here is inaccurate. We have rewritten it in revised manuscript. (see Page 1, lines 28-30)

Page 1, lines 28-30: "Our results provide new insights into the evolutionary behavior of the chromophoric and fluorescent properties of BB-BrC during O3 aging, which are of great significance for better understanding the heterogeneous oxidation pathways of BB-derived BrC in atmospheric environment."

2) Following p. 3 l. 11-29: When explaining that BB-BrC have been studied in the laboratory, the manuscript should connect to the work in the following seven papers by adding a short paragraph summarizing their findings: a) E.A. Pillar et al., Environ. Sci. Technol., 2017, 51, 4951-4959. b) J. Sun et al., Sun, Chem. Eng. J., 2019, 358, 456-466. c) E.A. Pillar et al., Environ. Sci. Technol., 2014, 48, 14352-14360. d) E.A. Pillar et al., J. Phys. Chem. A, 2015, 119, 10349-10359. e) A. Lavi, ACS Earth & Space Chem., 2017, 1 (10), 637-646. f) A.C.O. Magalhães, ACS Earth & Space Chem., 2017, 1, 353-360. g) Pillar-Little, Environments (2018), 5(9), 104.

It would be fundamentally important to connect the findings of the aromatic structures in the above seven papers with the material in the revision at multiple points of the manuscript such as p. 7 l. 29 and l. 34; p. 9 l. 10, l. 18, and l. 25; p. 10 l. 1; p. 11 l. 10 and l. 15.

Re: Thanks for the comments. We have carefully read these references. These studies provided some important insights of the $O_3$ oxidation mechanism of BB-derived oxy-aromatics (i.e. catechol and its substituted ones) occurred in air-water and air-solid interfaces (Lavi et al., 2017; Magalhães et al., 2017; Pillar et al., 2014, 2015, 2017; Pillar-Little and Guzman, 2018; Sun et al., 2019). They are helpful for us to investigate the evolutionary behaviors of light absorbing compounds during $O_3$ aging of BB-derived smoke compounds. According to the comments, we have revised the "Introduction" based on these literatures. (see Page 2, line 20, 24-25, 32; Page 3, lines

6-18)

In addition, these studies also revealed valuable information on the transformation of aromatic structures during $O_3$ oxidation, which are helpful for us to address the relevant statements on evolutionary behaviors of BB-BrC in this study. We have connected these findings with multiple points of the study and revised that in the current manuscript. (See Page 7, lines 13-16, 21-23; Page 9, lines 29-33; Page 11, lines 5-9, 14-18)

References:

[1] Lavi, A.; Lin, P.; Bhaduri, B.; Carmieli, R.; Laskin, A.; Rudich, Y., Characterization of Light-Absorbing Oligomers from Reactions of Phenolic Compounds and Fe(III). ACS Earth and Space Chemistry 2017, 1, (10), 637-646.

[2] Magalhães, A. C. O.; Esteves da Silva, J. C. G.; Pinto da Silva, L., Density Functional Theory Calculation of the Absorption Properties of Brown Carbon Chromophores Generated by Catechol Heterogeneous Ozonolysis. ACS Earth and Space Chemistry 2017, 1, (6), 353-360.

[3] Pillar, E. A.; Camm, R. C.; Guzman, M. I., Catechol Oxidation by Ozone and Hydroxyl Radicals at the Air–Water Interface. Environmental Science & Technology 2014, 48, (24), 14352-14360.

[4] Pillar, E. A.; Guzman, M. I., Oxidation of Substituted Catechols at the Air-Water Interface: Production of Carboxylic Acids, Quinones, and Polyphenols. Environ Sci Technol 2017, 51, (9), 4951-4959.

[5] Pillar, E. A.; Zhou, R.; Guzman, M. I., Heterogeneous Oxidation of Catechol. The Journal of Physical Chemistry A 2015, 119, (41), 10349-10359.

[6] Pillar-Little, E.; Guzman, M., An Overview of Dynamic Heterogeneous Oxidations in the Troposphere. Environments 2018, 5, (9).

[7] Sun, J.; Wei, B.; Mei, Q.; An, Z.; Wang, X.; He, M., Ozonation of 3-methylcatechol and 4-methylcatechol in the atmosphere and aqueous particles: Mechanism, kinetics and ecotoxicity assessment. Chemical Engineering Journal 2019, 358, 456-466.

3) p. 4 l. 38: What is the metal for the metal mesh holding the biomass materials? What is the temperature of the combustor through operation? Is it constant or variable? Indicate in the revised manuscript.

Re: In this study, the "metal" refers to "stainless steel", which was used to hold the biomass materials. The "metal mesh" has been revised to "stainless steel mesh". (see page 4, line 4)

In this study, the BB smoke samples were collected in a laboratory sampling system without any control conditions. The burning experiments simulated more likely a natural BB process. Each BB smoke filter sample was collected from a complete biomass burning process (from ignition to burnout). The temperature of the combustor is variable during the burning process. We have added this information of the sampling in the current manuscript. (see Page 4, lines 2-3)

4) p. 5 l. 1: What is the consideration for the gases adsorbed into the quartz filters?

Re: Thanks. Indeed, our quartz filters may adsorb gas-phase organic artifacts (i.e. semivolatile and intermediate volatility organic compounds) during BB smoke particles sampling (Geller et al., 2006; Parshintsev et al. 2011). According to previous studies, the average adsorbed organic artifacts (organic carbon, OC) amounts on quartz filters are very small (0.48-0.98 $\mu gC/cm^2$) (Arhami et al., 2006; Subramanian et al., 2004). In the current study, the OC contents on fresh BB smoke filters are in the range of ~250-750 $\mu gC/cm^2$, which are much higher than the possible artifacts. Therefore, the potential contributions from $O_3$ oxidation of possible artifacts on filters to bulk BB-BrC compounds can be neglected. We have added the relevant descriptions in the section 2.2 in revised manuscript. (see Page 4, lines 23-29)

References:

[1]  Arhami, M.; Kuhn, T.; Fine, P. M.; Delfino, R. J.; Sioutas, C., Effects of Sampling Artifacts and Operating Parameters on the Performance of a Semicontinuous Particulate Elemental Carbon/Organic Carbon Monitor. Environmental Science & Technology 2006, 40, (3), 945-954.

[2]  Geller, M. D.; Ntziachristos, L.; Mamakos, A.; Samaras, Z.; Schmitz, D. A.; Froines, J. R.; Sioutas, C., Physicochemical and redox characteristics of particulate matter (PM) emitted from gasoline and diesel passenger cars. Atmospheric Environment 2006, 40, (36), 6988-7004.

[3]  Parshintsev, J.; Ruiz-Jimenez, J.; Petäjä, T.; Hartonen, K.; Kulmala, M.; Riekkola, M.-L., Comparison of quartz and Teflon filters for simultaneous collection of size-separated ultrafine aerosol particles and gas-phase zero samples. Analytical and Bioanalytical Chemistry 2011, 400, (10), 3527-3535.

[4]  Subramanian, R.; Khlystov, A. Y.; Cabada, J. C.; Robinson, A. L., Positive and Negative Artifacts in Particulate Organic Carbon Measurements with Denuded and Undenuded Sampler Configurations Special Issue of Aerosol Science and Technology on Findings from the Fine Particulate Matter Supersites Program. Aerosol Science and Technology 2004, 38, (sup1), 27-48.

L. 10: glass garden? Do you mean a terrarium?

Re: Yes, the "glass garden" is a terrarium, which refers to a glass dish. It was used to spread the filter samples exposed in ozone environment. The term "glass garden" has been revised to "glass dish ($\Phi$ = 90 mm)" in the text. (see Page 4, line 16)

L. 13: Why is the ozone so high (70 ppm) and how is it environmentally relevant? Why is the relative humidity fixed at 40%? Explain in the revised manuscript.

Re: Thanks for your comments. In this study, the experiment conditions (e.g., $O_3$ concentrations, humidity, etc.) of $O_3$ aging process were applied mainly based on the results of $O_3$ simulation experiments in some previous studies. In fact, many $O_3$

oxidation simulation experiments have been conducted to investigate the aging of carbonaceous compounds under different $O_3$ concentration (20 ppb – 12,200 ppm) (Baduel et al., 2011; D'Anna et al., 2009; Pillar et al., 2014, 2017). For example, low $O_3$ concentrations (20 ppb - 6 ppm) had been used for oxidizing the thin films of humic matters (Baduel et al., 2011; D'Anna et al., 2009) and oxy-aromatics (i.e. catechol and its substituted ones) (Pillar et al., 2014, 2017), in which the changes of the uptake coefficient of $O_3$ and the early aging mechanism occurred in air-particle interface had been explored. However, to explore the changes of physicochemical properties of particulate samples from combustion process and secondary chemical reactions, or to investigate the formation of light absorbing organic compounds during $O_3$ aging process, a relatively high $O_3$ concentrations (20 ppm-12,200 ppm) were generally used in the simulation experiments (Li et al., 2013, 2015; Decesari et al., 2002; Gallimore et al., 2011; Pillar et al., 2015; Zhu et al., 2019). Moreover, some studies have revealed that the oxidation mechanism at higher $O_3$ concentration is similar to that done under much lower $O_3$ concentration (Pillar et al., 2015; Gallimore et al., 2011). In this study, the main objective is to investigate the evolutionary behavior of chromophoric BrC compounds during $O_3$ aging of BB smoke samples, thus the aging simulation experiment was conducted at a relative higher $O_3$ concentration (70 ppm). The detailed explanation for high $O_3$ concentration used in this study have been provided in "S1. Ozone aging reactor and operation" in revised supporting information (SI). (see Page 1, lines 13-34 in revised SI)

Moreover, in this study, the $O_3$ exposure amount for 1 h in reactor were $\sim 1.7 \times 10^{15}$ molec $cm^{-3}$ h. For a highly $O_3$ polluted ($\sim$120 ppb) area (Chen et al., 2020), 24 h-average atmospheric $O_3$ exposure amount were $\sim 7.1 \times 10^{13}$ molec $cm^{-3}$ h. In this case, the oxidation for 1 h in our reactor was approximately equivalent to 260 d of oxidation at polluted atmosphere. However, the smoke samples of this study were highly condensed and coagulated on the filter, so the exposure area of particles were greatly reduced. As a result, the equivalent day for $O_3$ oxidation should be highly shortened. Importantly, our results demonstrated many similar oxidation behaviors of organic chromophores to

those of atmospheric humic matter and BB-derived oxy-aromatics under low $O_3$ concentration (20 ppb- 6 ppm) (Baduel et al., 2011; D'Anna et al., 2009; Pillar et al., 2014, 2017). Therefore, we believed that the evolutionary behaviors of BB-BrC revealed here should be similar to those occurred under atmospheric relevant $O_3$ concentrations during their lifetime in atmosphere. Some descriptions on atmospheric relevance have been added in section S1 in revised supporting information. (See Page 1, line 37 to Page 2, line 12 in revised SI).

In addition, many previous studies have revealed that relative humidity (RH) has some effects on the oxidation of organic compounds, in which the rate of reaction generally increased with RH increase (Baduel et al., 2011; Gallimore et al., 2011; Pillar et al., 2015). However, the objectives of this study are mainly to investigate the $O_3$ aging behavior of BrC in BB smoke samples, and to explore the influences of the type of fuels and oxidation time on the evolutionary behavior of light absorbing BrC components. Therefore, a moderate RH (~40%) was used in this study. Certainly, much more studies on $O_3$ aging of BB emission as a function of $O_3$ concentration and RH should be conducted in the future work. We have added some descriptions to state that in revised manuscript. (see Page 13, lines 29-33)

References:

[1] Baduel, C.; Monge, M. E.; Voisin, D.; Jaffrezo, J.-L.; George, C.; Haddad, I. E.; Marchand, N.; D'Anna, B., Oxidation of Atmospheric Humic Like Substances by Ozone: A Kinetic and Structural Analysis Approach. Environmental Science & Technology 2011, 45, (12), 5238-5244.

[2] Chen, Z.; Li, R.; Chen, D.; Zhuang, Y.; Gao, B.; Yang, L.; Li, M., Understanding the causal influence of major meteorological factors on ground ozone concentrations across China. Journal of Cleaner Production 2020, 242, 118498.

[3] D'Anna, B.; Jammoul, A.; George, C.; Stemmler, K.; Fahrni, S.; Ammann, M.; Wisthaler, A., Light-induced ozone depletion by humic acid films and submicron aerosol particles. Journal of Geophysical Research 2009, 114, (D12).

[4]  Decesari, S.; Facchini, M. C.; Matta, E.; Mircea, M.; Fuzzi, S.; Chughtai, A. R.; Smith, D. M., Water soluble organic compounds formed by oxidation of soot. Atmospheric Environment 2002, 36, (11), 1827-1832.

[5]  Gallimore, P. J.; Achakulwisut, P.; Pope, F. D.; Davies, J. F.; Spring, D. R.; Kalberer, M., Importance of relative humidity in the oxidative ageing of organic aerosols: case study of the ozonolysis of maleic acid aerosol. Atmos. Chem. Phys. 2011, 11, (23), 12181-12195.

[6]  Li, Q.; Shang, J.; Zhu, T., Physicochemical characteristics and toxic effects of ozone-oxidized black carbon particles. Atmospheric Environment 2013, 81, 68-75.

[7]  Li, Q.; Shang, J.; Liu, J.; Xu, W.; Feng, X.; Li, R.; Zhu, T., Physicochemical characteristics, oxidative capacities and cytotoxicities of sulfate-coated, 1,4-NQ-coated and ozone-aged black carbon particles. Atmospheric Research 2015, 153, 535-542.

[8]  Pillar, E. A.; Camm, R. C.; Guzman, M. I., Catechol Oxidation by Ozone and Hydroxyl Radicals at the Air–Water Interface. Environmental Science & Technology 2014, 48, (24), 14352-14360.

[9]  Pillar, E. A.; Guzman, M. I., Oxidation of Substituted Catechols at the Air-Water Interface: Production of Carboxylic Acids, Quinones, and Polyphenols. Environ Sci Technol 2017, 51, (9), 4951-4959.

[10] Pillar, E. A.; Zhou, R.; Guzman, M. I., Heterogeneous Oxidation of Catechol. The Journal of Physical Chemistry A 2015, 119, (41), 10349-10359.

[11] Zhu, J.; Chen, Y.; Shang, J.; Zhu, T., Effects of air/fuel ratio and ozone aging on physicochemical properties and oxidative potential of soot particles. Chemosphere 2019, 220, 883-891.

5) p. 6 l. 2-3: Despite mentioning that more details are in the SI, provide literature references for the concepts of SUVA254, AAE, MAE365, and HIX.

Re: thanks. Some necessary literature references for the concepts of SUVA254, AAE, MAE365, and HIX have been added in the revised text.

6) p. 6 l. 24: Provide software version and company name that sell it.

Re: The software version and company name are OriginPro 2018C and OriginLab (USA), respectively, which have been provided in revised text. (see Page 5, line 39)

7) p. 7 l. 37: Despite what is explained, it appears that in this work there is no soot. How is this point connected to the results?

Re: Thanks for the comment. Soot particles often refer to elemental carbon (EC), black carbon (BC) or even light-absorbing carbons in atmosphere (Li et al., 2013, 2015; Han et al., 2013), which are largely formed by incomplete combustion of biomass and fossil fuels (Khalizov et al., 2010). As reported in previous studies and measured in this study, biomass burning could release large amounts of soot, which contributed to 6-55% of total mass of particles (Hong et al., 2017; Schmidl et al, 2011). Moreover, many previous studies have revealed that $O_3$ oxidation of soot particles could form new chromophoric compounds (Li et al., 2013, 2015; Decesari et al., 2002; Zhu et al., 2019), suggesting that it is one of the important reactions during the aging of BB BrC. Therefore, the $O_3$ oxidation of soot component in smoke samples was discussed in this study. For better understanding, we have made some descriptions in the revised supporting information. (see page 1, lines 27-31 in SI)

References:
[1] Decesari, S.; Facchini, M. C.; Matta, E.; Mircea, M.; Fuzzi, S.; Chughtai, A. R.; Smith, D. M., Water soluble organic compounds formed by oxidation of soot. Atmospheric Environment 2002, 36, (11), 1827-1832.
[2] Han, C.; Liu, Y.; He, H., Heterogeneous photochemical aging of soot by NO2 under simulated sunlight. Atmospheric Environment 2013, 64, 270-276.
[3] Hong, L.; Liu, G.; Zhou, L.; Li, J.; Xu, H.; Wu, D., Emission of organic carbon, elemental carbon and water-soluble ions from crop straw burning under flaming

and smoldering conditions. Particuology 2017, 31, 181-190.

[4] Khalizov, A. F.; Cruz-Quinones, M.; Zhang, R., Heterogeneous Reaction of NO2 on Fresh and Coated Soot Surfaces. The Journal of Physical Chemistry A 2010, 114, (28), 7516-7524.

[5] Li, Q.; Shang, J.; Zhu, T., Physicochemical characteristics and toxic effects of ozone-oxidized black carbon particles. Atmospheric Environment 2013, 81, 68-75.

[6] Li, Q.; Shang, J.; Liu, J.; Xu, W.; Feng, X.; Li, R.; Zhu, T., Physicochemical characteristics, oxidative capacities and cytotoxicities of sulfate-coated, 1,4-NQ-coated and ozone-aged black carbon particles. Atmospheric Research 2015, 153, 535-542.

[7] Schmidl, C.; Luisser, M.; Padouvas, E.; Lasselsberger, L.; Rzaca, M.; Ramirez-Santa Cruz, C.; Handler, M.; Peng, G.; Bauer, H.; Puxbaum, H., Particulate and gaseous emissions from manually and automatically fired small scale combustion systems. Atmospheric Environment 2011, 45, (39), 7443-7454.

[8] Zhu, J.; Chen, Y.; Shang, J.; Zhu, T., Effects of air/fuel ratio and ozone aging on physicochemical properties and oxidative potential of soot particles. Chemosphere 2019, 220, 883-891.

8) p. 7 l. 12-24; p. 8 l. 12-13; and p. 13 l. 21: Changes in absorption with wavelength for chromophores and/or fluorophores, and the concept of photobleaching, have been investigated in the following seminal papers that introduced the ideas and should be discussed here with the addition of a few relevant statements: a) A.G. Rincon at al., J. Phys. Chem. A, 2009, 113, 10512-10520. b) A.G. Rincon at al., J. Phys. Chem. Lett., 2010, 1, 368-373. c) A.J. Eugene et al., J. Phys. Chem. A, 2016, 120, 3817-3826 d) S.-S. Xia et al., J. Phys. Chem. A, 2018, 122, 6457-6466.

Re: Thanks. We have carefully read these references (Eugene et al., 2016; Rincón et al., 2009, 2010; Xia et al., 2018). In these studies, the changes in absorption with wavelength for chromophores and/or fluorophores during photochemical reaction of BB-derived organic compounds (i.e. glyoxylic acid, pyruvic acid, phenolic compounds)

were investigated, and the photobleaching behaviors for the chromophores were also discussed. These findings are very helpful for us to explain the bleaching behaviors of BB BrC, especial for those associated to changes in absorption depending on wavelength, during $O_3$ oxidation. According to comments, we have added some descriptions and revised that in the current manuscript. (Page 6, lines 35-39; Page 7, lines 20-23; Page 8, lines 2-4; Page 9, lines 29-33)

References:

[1]  Eugene, A. J.; Xia, S.-S.; Guzman, M. I., Aqueous Photochemistry of Glyoxylic Acid. The Journal of Physical Chemistry A 2016, 120, (21), 3817-3826.

[2]  Rincón, A. G.; Guzmán, M. I.; Hoffmann, M. R.; Colussi, A. J., Optical Absorptivity versus Molecular Composition of Model Organic Aerosol Matter. The Journal of Physical Chemistry A 2009, 113, (39), 10512-10520.

[3]  Rincón, A. G.; Guzmán, M. I.; Hoffmann, M. R.; Colussi, A. J., Thermochromism of Model Organic Aerosol Matter. The Journal of Physical Chemistry Letters 2010, 1, (1), 368-373.

[4]  Xia, S.-S.; Eugene, A. J.; Guzman, M. I., Cross Photoreaction of Glyoxylic and Pyruvic Acids in Model Aqueous Aerosol. The Journal of Physical Chemistry A 2018, 122, (31), 6457-6466.

9) p. 7 l. 34: What is the meaning of polycondensation? Clarify.

Re: Thanks. The "polycondensation" means that some highly stable and condensed chromophores were formed during $O_3$ oxidation. These chromophores might lead to the high HIX values of aged BrC. In addition, the polyhydroxylation of aromatic compounds might also lead to high HIX values for chromophores (Pillar et al., 2014, 2015, 2017), so that the "polycondensation" might be inaccurate. We have deleted this sentence and added some other descriptions in the current manuscript. (see Page 7, lines 16-23)

Page 7, lines 16-23: "The noticeable HIX increases seen for the three types of BB-BrC indicate that the $O_3$ aging may strongly decompose the protein-like fluorophores, probably phenolic compounds (Chen et al., 2016a), to form polyhydroxylated aromatic species or newly humic-like fluorophores (Pillar et al., 2014, 2015, 2017; Decesari et al., 2002; Li et al., 2013) (Figure 3b). For example, $O_3$ oxidation of phenolic compounds could form polyhydroxylated aromatic compounds with absorption red-shift, which might lead to their HIX values increase (Lavi et al., 2017; Magalhães et al., 2017; Pillar et al., 2015; Rincón et al., 2009, 2010)."

10) p. 30 l. 30: ". . .is in good. . ."

Re: Revised.